# Effect of a Herbal Therapy on Clinical Symptoms of Acute Lower Uncomplicated Urinary Tract Infections in Women: Secondary Analysis from a Randomized Controlled Trial

**DOI:** 10.3390/antibiotics8040256

**Published:** 2019-12-07

**Authors:** Winfried Vahlensieck, Horst Lorenz, Anne Schumacher-Stimpfl, Roland Fischer, Kurt G. Naber

**Affiliations:** 1Department of Urology, Kurpark-Klinik, Kurstraße 41-45, 61231 Bad Nauheim, Germany; 2BBS—Büro für Biometrie und Statistik, Im Unterfeld 17, 63543 Neuberg, Germany; horst.lorenz@bbs-neuberg.de; 3Medice Arzneimittel Pütter GmbH & Co. KG, Kuhloweg 37, 58638 Iserlohn, Germany; a.schumacher@medice.de (A.S.-S.); r.fischer@medice.de (R.F.); 4Department of Urology, Technical University of Munich, Urology, Ismaninger Str. 22, 81675 München, Germany

**Keywords:** urinary tract infection, symptomatic treatment, herbal treatment, patient-reported outcomes, Acute Cystitis Symptom Score, ACSS

## Abstract

**Background**: Antibiotics are commonly used as first-line treatment for acute lower uncomplicated urinary tract infections (uUTIs). However, antimicrobial resistance is a growing global problem and efficacious nonantibiotic treatment options are urgently needed. Methods: A secondary analysis was conducted with data from a randomized, controlled, double-blind trial comparing a fixed combination of extracts of restharrow root, Java tea, and goldenrod herb (Aqualibra^®^) to placebo in 200 women with acute lower uUTI. Symptom scores reported in the original trial were reanalyzed and adjusted to the definitions of the Acute Cystitis Symptom Score (ACSS). Results: Based on a subgroup of patients with evaluable microbiologic data (*n* = 122), the decrease of the mean sum-score of three typical ACSS-adjusted symptoms showed significant superiority of the herbal preparation over placebo already after one day of treatment (*p* = 0.0086); on Day 7, the average difference was −1.9 score points (*p* < 0.0001). The superior efficacy of the herbal preparation on Day 1 was mainly driven by a difference in response rates of the symptom ‘dysuria’ (group difference: −29.4%, *p* = 0.0042). Furthermore, significantly fewer patients in the verum group required antibiotic therapy (15.3% vs. 49.2%, *p* = 0.0001). These results were confirmed in the intention-to-treat (ITT) population (*n* = 200). Conclusions: A fixed combination of extracts of restharrow root, Java tea, and goldenrod herb was superior to placebo regarding symptom relief and prevention of antibiotic use in women with lower uUTI. Trial registration: ClinicalTrials.gov: NCT04032574.

## 1. Introduction

Urinary tract infections (UTIs) are among the most common infectious diseases in clinical practice, and of these, most are classified as uncomplicated UTIs (uUTIs) [1]. Current guidelines recommend antibiotics as first-line treatment for acute uUTIs [2]. However, antibiotic use is recognized as one of the main drivers of antimicrobial resistance, which has become a rapidly growing global problem [3].

The most common pathogens of uUTI in women are uropathogenic *Escherichia coli* (UPEC), accounting for up to 80% of cases [1]. The worldwide antibiotic resistance rates of *E. coli* in nosocomial UTIs were reported to exceed 25% for most common antibiotics [4]. Susceptibility of *E. coli* to fosfomycin, nitrofurantoin, and mecillinam still remains high [5,6]. 

Reduction of antibiotic prescribing for uUTIs is crucial to reduce the development of antimicrobial resistance. Accordingly, recent national guidelines emphasize the need to avoid inappropriate antibiotic prescribing [7]. The symptomatic treatment has been explored as an alternative treatment option for uUTI. Studies investigating the efficacy of nonsteroidal anti-inflammatory drugs (NSAIDs) compared to antibiotics showed promising results, although NSAIDs were inferior to antibiotics for symptom relief and seemed to be associated with an increased risk of pyelonephritis [8,9,10]. In light of these findings, symptomatic treatment alone may be considered for acute uncomplicated cystitis with mild to moderate symptoms [7].

Phytotherapy is another potential alternative treatment strategy for uUTIs; however, the currently available evidence for herbal interventions is not well established yet. The ongoing REGATTA trial is currently comparing the effects of bearberry extract (Arctuvan^®^) to fosfomycin in women with uUTI. The aim of the study is to investigate whether initial treatment with bearberry extract reduces the number of antibiotic courses without significantly increasing symptom burden. The results of this study are still pending [11]. Wagenlehner et al. recently reported that a combination of centaury powder, lovage root powder, and rosemary leaf powder (Canephron^®^ N) was noninferior to fosfomycin in treating acute lower uUTIs, as measured by the proportion of patients who received an additional antibiotic treatment within 38 days [12].

Aqualibra^®^ is an approved herbal medicinal product containing a combination of extracts of restharrow root (*Ononidis radix*), Java tea (*Orthosiphonis folium*), and goldenrod herb (*Solidaginis herba*). All three herbal substances are monographed by the Committee on Herbal Medicinal Products (HMPC) of the European Medicines Agency (EMA) and intended for the treatment of uUTI. The respective assessment reports summarize available data on their pharmacological characteristics, which include diuretic, anti-inflammatory, antibacterial, and spasmolytic activities [13,14,15]. Furthermore, *Orthosiphon* leaf extract has anti-adhesive activity in vitro and in vivo and reduces the expression of virulence factors of uropathogenic *E. coli* [16,17].

The efficacy and safety of the herbal medicinal product were investigated in an open-label, noninterventional study in 1904 patients with lower uUTI. After 7 days of treatment, 78% of patients were completely free of all symptoms of acute cystitis (dysuria, urination frequency [pollakisuria], urgency, and suprapubic pain), while 11% still had one mild symptom. In 9% of the patients, the symptoms worsened and antibiotic therapy was administered; in 2% of the patients, the symptoms remained unchanged [18].

In addition, a randomized, placebo-controlled, double-blind phase III trial was conducted in the early 1990s. This study investigated the efficacy and safety of the herbal medicinal product in 200 women with acute lower uUTI. The primary endpoint was microbiologic response (reduction in bacterial urine culture counts by at least 10^2^ colony forming units (CFU)/mL), with a responder rate of 64.4% in the verum group and 25.4% in the placebo group (*p* < 0.0001) [19].

Current guidelines on clinical trial designs have emphasized the importance of including clinical efficacy parameters that focus on patient-reported outcomes, i.e., improvement of symptoms [20,21].

Therefore, a secondary individual patient data (IPD) analysis of the randomized controlled trial was performed, focusing on symptomatic rather than microbiologic outcomes.

To this end, new symptomatic endpoints based on the Acute Cystitis Symptom Score (ACSS) were defined. The ACSS is a validated, standardized self-reporting questionnaire used to evaluate the symptoms of acute uncomplicated cystitis in women, which can be used for clinical diagnostics and patient-reported outcome assessment [22,23,24,25]. In addition, the proportion of patients who required additional antibiotic therapy to treat their acute lower uUTIs was analyzed more closely.

The aim of this re-analysis was to evaluate whether the herbal combination was superior to placebo in the improvement of patient-relevant clinical parameters and whether it prevented additional antibiotic usage.

## 2. Results

Of the 200 enrolled patients, 98 were randomized to the herbal combination and 102 were randomized to placebo. Twenty-one patients did not fulfill the inclusion/exclusion criteria, i.e., they had bacterial counts <10^4^ CFU/mL (*n* = 17) or used prohibited concomitant medication (*n* = 4). In addition, a high number of patients (*n* = 57) had so-called contaminated urine cultures (e.g., cultures with more than one organism at baseline or on Day 7) and were thus classified as drop-outs from microbiological evaluation (Appendix A). The original analysis of the primary endpoint was based only on patients with evaluable microbiologic data (*n* = 122; microPP population).

To confirm the findings of the original analysis using modern endpoints, the microbiologic data were re-analyzed using microbiologic response criteria as defined by current guidelines [20,21], i.e., a bacterial count reduction from ≥10^5^ to <10^3^ CFU/mL. According to this response definition, the microbiologic response rate in the microPP population was 62.7% in the verum group and 25.4% in the placebo group (*p* < 0.0001) (Appendix A).

The present secondary analysis of the clinical symptoms using the ACSS-adjusted score was performed for both the intention-to-treat (ITT) (*n* = 200) and microPP population.

The reduction of the mean sum-scores of the ACSS-adjusted three ‘typical symptoms’ showed significant superiority of the herbal combination over placebo already after one day of treatment in both the microPP (*p* = 0.0086) and ITT (*p* = 0.0081) population. By the end of treatment (Day 7), the mean sum-scores of the ACSS-adjusted items were decreased from baseline in both the verum and placebo group, while the difference between verum and placebo became more pronounced (microPP: −1.9, *p* < 0.0001; ITT: −1.1, *p* = 0.0008) (Figure 1, Table 1).

The determination of global scores (O’Brien) and maximum likelihood estimators (linear mixed model) confirmed these findings. Additional analyses of clinical symptoms based on the originally documented symptoms (original items) showed a very similar pattern and, therefore, gave further validation to the results (Appendix A). Furthermore, the response rates for individual symptoms were analyzed. Responders were defined as patients whose score for an individual ACSS-adjusted symptom improved by at least one point. Analysis of responder rates for individual ACSS-adjusted symptoms revealed that the superior efficacy of the herbal combination on Day 1 was mainly due to an improvement in the symptom ‘dysuria’; the difference between the response rates for ‘dysuria’ between the groups was 29.4% (*p* = 0.0042) in the microPP population and 19.0% (*p* = 0.0111) in the ITT population (Figure 2). A trend for superiority of the herbal combination for the symptom urination frequency (‘pollakisuria’) was observed on Day 1. The difference between the groups became significant on Day 7, with a difference of 29.8% in the microPP population (*p* < 0.0001) and a difference of 16.4% in the ITT population (*p* = 0.0027) (Figure 2). A statistically significant superiority of the herbal combination regarding the response rate for the symptom ‘suprapubic pain’ was observed only on Day 1, both in the microPP (*p* = 0.0249) and ITT (*p* = 0.0325) population (Figure 2).

The proportion of so-called “clinically cured” patients (defined as patients who had a total ACSS-adjusted score below 3 and no individual ACSS-adjusted symptom score above 1) on Day 7 was significantly higher in the verum group compared to the placebo group (31.1% vs. 14.7%, *p* = 0.0001), based on the microPP population. A statistically significant difference in cure rates was also seen in the ITT population (*p* = 0.0106) (Figure 3).

The proportion of patients who were prescribed antibiotic therapy during the study (Day 1 or Day 7) was significantly reduced in the verum vs. the placebo group (15.3% vs. 49.2%, *p* = 0.0001) when based on the microPP population. The difference was also significant in the ITT population (*p* = 0.0248) (Figure 4).

## 3. Discussion

The present retrospective secondary analysis shows that the herbal combination of dry extracts of restharrow root (*Ononidis radix*), Java tea (*Orthosiphonis folium*), and goldenrod herb (*Solidaginis herba*) is superior to placebo for the improvement of clinical symptoms of uUTI. Thus, in addition to achieving a microbiologic response, as shown in the original analysis, the herbal combination also achieves clinical response as measured using a modern validated symptom score that was not available at the time of the original trial.

Remarkably, a significant difference between treatment groups regarding the overall improvement of clinical symptoms (sum-score of ACSS-adjusted items) was seen as early as one day after treatment initiation. This difference was mainly due to differences in response rates for the symptoms ‘dysuria’ and ‘suprapubic pain’. The effect on urination frequency (‘pollakisuria’) was delayed, with significant differences evident only on Day 7, possibly because patients were recommended to increase their liquid intake during the study. 

In addition, the proportion of “clinically cured” patients on Day 7 was significantly higher in the verum group compared to the placebo group. The definition of clinical cure as used in this post-hoc analysis is however rather strict and was adjusted from an earlier analysis, in which Alidjanov et al. [23] used a sum-score of 4 or less including all six ACSS symptoms with no individual symptom score above 1 (mild).

This secondary analysis was based on the ITT population (*n* = 200) in addition to the microPP population (*n* = 122). Because the ITT population includes all patients who were randomized, the data for analysis include patients whose urine bacterial counts were too low or not evaluable at the beginning of the study. The effects of the herbal combination regarding clinical symptom reduction were comparable between the two populations, suggesting that the standard microbiologic definition of acute cystitis with a bacterial count of at least 10^5^ CFU/mL excludes many patients who might benefit from symptomatic treatment.

The herbal combination was also superior to placebo in preventing antibiotic treatment. This finding is particularly relevant in light of the high antibiotic prescribing rates for uUTI in the community setting. Uncomplicated UTI is one of the most common bacterial infections worldwide [1]. Current treatment guidelines recommend antibiotics as first-line treatment for uUTIs [2]. Thus, high antibiotic prescribing rates for uUTIs may contribute to the global increase in antimicrobial resistance.

The identification of alternative, nonantibiotic treatment options for uUTI is crucial. Promising results have been obtained in studies investigating the use of NSAIDs and herbal preparations [8,9]. Wagenlehner et al. recently investigated a combination of powdered centaury herb, lovage root, and rosemary leaves for the treatment of lower uUTI and showed that 83.5% of patients in the per-protocol population did not require further antibiotic therapy within 38 days [12]. Similarly, the current analysis shows that most patients in the verum group (84.7% in the microPP population) did not require additional antibiotic therapy during the study (7 days). The relatively high proportion of patients in the placebo group that did not require additional antibiotic therapy might be attributed to the self-limiting character of uUTIs associated with high spontaneous cure rates.

The findings of this secondary analysis suggest that symptomatic treatment with a combination of dry extracts of restharrow root, Java tea, and goldenrod herb can contribute to a reduction of antibiotic prescribing rates for lower uUTIs. This hypothesis should be further investigated in clinical trials that include sufficient follow-up periods after the end of treatment. Furthermore, based on current guideline recommendations, head-to-head studies comparing the herbal combination to the current standard of care treatment (i.e., antibiotic therapy) should be conducted. Previous studies have shown that NSAIDs are clinically inferior to antibiotics in the treatment of UTI [8,9]. Thus, it would be interesting to investigate in future studies whether the herbal combination with a broad pharmacological spectrum is noninferior to antibiotics in terms of symptom relief.

## 4. Material and Methods

### 4.1. Trial Design

This is a secondary analysis of a randomized, placebo-controlled, parallel-group, double-blind phase III trial comparing a herbal combination to placebo in women with acute lower uUTI. The trial was performed between October 1991 and March 1992. The study design and methods have been described previously [19] and are summarized below.

### 4.2. Patients

Women aged 18–75 years were enrolled if they met the following inclusion criteria: acute lower uUTI occurring for the first time or acute relapse of recurrent uUTI, typical symptoms of cystitis (pollakisuria, dysuria, and urgency), bacterial count of 10^4^–10^6^ CFU/mL in midstream urine, >20 leukocytes/µL of urine, and no antibiotic treatment required according to the investigator. Exclusion criteria included: antibiotic treatment during the past 8 days, use of concomitant medication that may influence the UTI, any signs of complicated UTI or pyelonephritis, other infections (trichomoniasis, chlamydiosis, gonorrhea). Pregnant or nursing women or women not using highly effective methods of contraception were also excluded. All patients provided written informed consent to participate. A complete list of inclusion and exclusion criteria is provided in Appendix A. 

### 4.3. Randomization and Masking

Patients were randomized 1:1 to either verum or placebo, according to a computer-generated randomization sequence with a block size of ten. Patients and investigators were blinded to treatment allocation.

### 4.4. Procedures

The verum and placebo medication was administered orally as two coated tablets, three times a day, for seven days. One coated tablet of Aqualibra^®^ (manufactured by Medice Arzneimittel Pütter GmbH & Co. KG) contains the dry extracts of restharrow root (*Ononidis radix*, 80 mg), Java tea (*Orthosiphonis folium*, 90 mg), and goldenrod herb (*Solidaginis herba*, 180 mg). Patients were advised to take the medication with plenty of liquids evenly distributed throughout the day, at least 1 hour before or after meals.

The study protocol included a total of three visits: a baseline visit (Day 0, treatment initiation) and two visits after 1 (Day 1) and 6 ± 1 (Day 7) days of treatment, respectively. Urine bacterial count was determined on Days 0 and 7; clinical symptoms were assessed at all three study visits. A full list of assessments conducted at each study visit is provided in Appendix A.

### 4.5. Outcomes

The objective of the trial was to compare the efficacy and safety of the herbal combination vs. placebo in the treatment of acute lower uUTI in women.

In the original analysis of this study, a microbiologic outcome was used for the assessment of efficacy: the primary endpoint was the response rate, defined as the percentage of patients with a reduction in bacterial urine culture counts by at least 10^2^ CFU/mL at the final visit. This primary analysis was performed on the subgroup of patients with evaluable microbiologic data (microPP population, *n* = 122). This means that the following patients were excluded from microPP:patients whose bacterial count was less than 10^4^ CFU/mL at the time of inclusionpatients whose urine cultures were contaminated and, therefore, not evaluablepatients who took concomitant medication that could influence the results of the urine cultures. For this secondary analysis, we defined a new patient-relevant clinical endpoint (i.e., clinical improvement of symptoms) based on the Acute Cystitis Symptom Score (ACSS) (Appendix A), a validated, standardized self-reporting questionnaire used to evaluate the symptoms of acute uncomplicated cystitis in women [22,23,24,25,26]. In addition, we analyzed whether the herbal combination reduced the rate of antibiotic use.

The secondary analysis was performed for both the ITT population (all randomized patients, *n* = 200) and the microPP population (*n* = 122).

### 4.6. Statistical Analysis

The ACSS consists of 18 questions categorized into four domains: ‘typical symptoms’ (*n* = 6), ‘differential diagnosis’ (*n* = 4), ‘quality of life’ (QOL) (*n* = 3), and ‘additional questions for underlying conditions’ (*n* = 5). Each item in the ‘typical’, ‘differential’, and ‘QOL’ domain has a Likert-type response scale ranging from 0 (no symptom) to 3 (severe), while the ‘additional’ domain uses yes/no questions.

The ACSS domain ‘typical symptoms’ includes the symptoms urination frequency (pollakisuria), urgency, dysuria, feeling of incomplete bladder emptying after voiding, suprapubic pain, and visible gross hematuria. The originally reported symptoms ‘pain when urinating’ and ‘burning when urinating’ were combined to represent the ACSS symptom ‘dysuria’. The originally reported symptoms ‘frequency of urgency during the day’, ‘frequency of urgency during the night’ and ‘frequency of urgency during day and night’ were combined to represent the ACSS symptom ‘pollakisuria’. The originally reported symptom ‘suprapubic pain’ is equivalent to the ACSS symptom ‘suprapubic pain’.

Thus, from the reported symptoms, three so-called ACSS-symptoms could be adapted and determined according to Alidjanov et al. [23] (symptom 1 = ‘pollakisuria’, symptom 3 = ‘dysuria’, and symptom 5 = ‘suprapubic pain’).

All reported symptoms (documented on 4-point scales: 0 = no, 1 = mild, 2 = moderate, 3 = severe) were descriptively evaluated (patients with improvements of at least one score point) and compared between treatment groups at each visit (baseline, Day 1 and Day 7) applying Wilcoxon’s 2-sample test and Fisher’s exact test, respectively, in accordance to the qualitative nature of the variables.

The score totals were investigated and compared for treatment groups applying 2-sample t-tests and assuming approximately normally distributed data. In addition, global scores and maximum likelihood estimators (linear mixed model) were determined to perform a sensitivity analysis and to investigate the correlation pattern between time points [27]. Finally, the number of patients who received antibiotics during the study was evaluated (chi-squared-test [28]).

All analyses were performed for the ITT population (*n* = 200) and repeated for a subgroup of patients with evaluable microbiologic data, the so-called microPP population (*n* = 122). The homogeneity of the ITT population and the microPP population was descriptively investigated in the clinical study report. 

The statistical calculations and plots were carried out applying the R-software [29]. Significance tests were determined and interpreted on a descriptive basis. In total, an exploratory analysis [27] was performed.

### 4.7. Ethics Approval and Consent to Participate

The randomized controlled trial was conducted according to the Declaration of Helsinki. The trial protocol (Study code 97.019/91) was reviewed and approved by an independent Ethics Committee (Landesärztekammer Baden-Württemberg). All participants provided written informed consent to participate before enrollment.

### 4.8. Consent for Publication

All participants provided written informed consent for the publication of the data before enrollment.

### 4.9. Availability of Data and Material

The datasets used and/or analyzed during the current study are available from the corresponding author on reasonable request.

## 5. Conclusions

In conclusion, the presented study with additional post-hoc analysis shows that the herbal combination is superior to placebo in terms of symptom relief and prevention of antibiotic use in women with lower uUTI. Thus, symptomatic treatment with the herbal combination represents a potential therapeutic alternative to antibiotic therapy and should be further investigated in clinical trials.

## Figures and Tables

**Figure 1 antibiotics-08-00256-f001:**
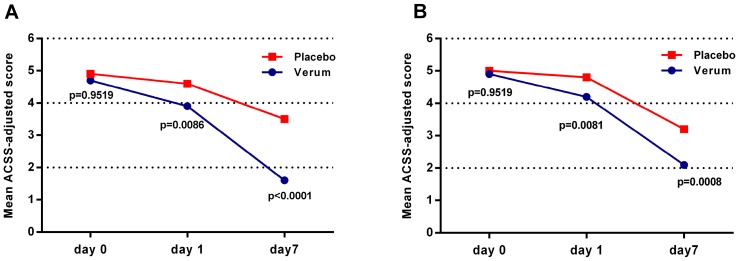
Comparison of mean sum-scores of the Acute Cystitis Symptom Score (ACSS)-adjusted ‘typical domain’ symptoms from Day 0 to Day 7 in the (**A**) population of patients with evaluable microbiologic data (microPP) and (**B**) intention-to-treat (ITT) population (Wilcoxon’s 2-sample test).

**Figure 2 antibiotics-08-00256-f002:**
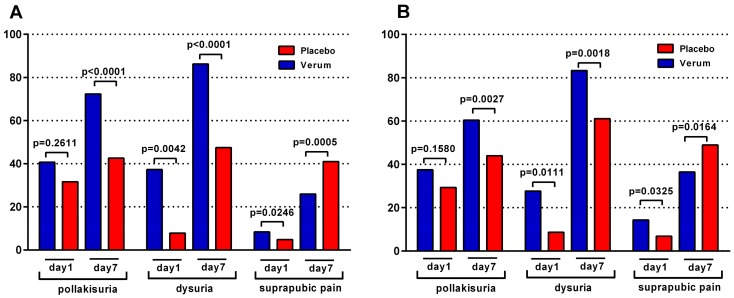
Proportion of patients achieving improvement by at least one point for an individual ACSS-adjusted symptom in the (**A**) microPP and (**B**) ITT population (chi-squared test).

**Figure 3 antibiotics-08-00256-f003:**
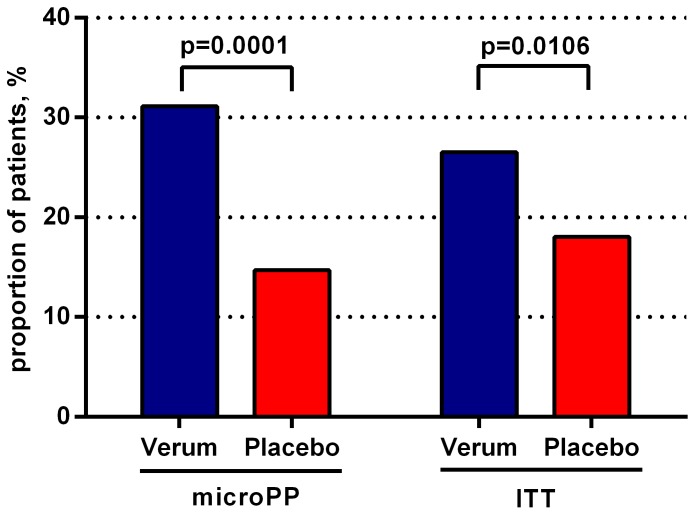
Proportion of “clinically cured” patients (defined as patients who had a total ACSS-adjusted score below 3 and no individual ACSS-adjusted symptom score above 1) on Day 7 in the microPP and ITT population (chi-squared test).

**Figure 4 antibiotics-08-00256-f004:**
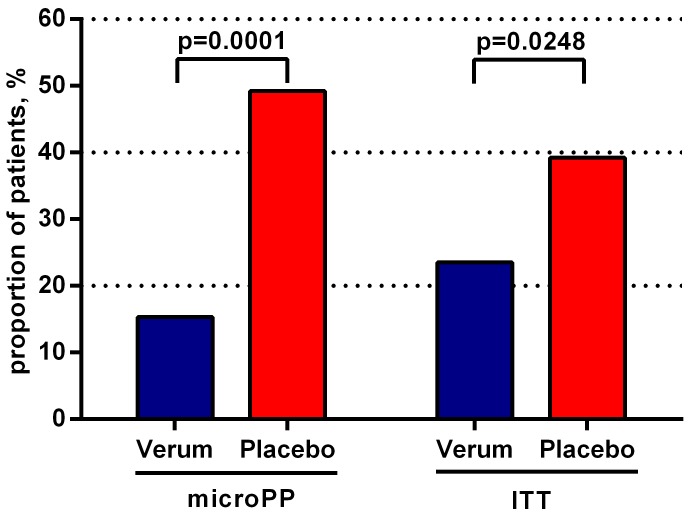
Proportion of patients who were prescribed additional antibiotic therapy on Day 1 or Day 7 in the microPP and ITT population (chi-squared test).

**Table 1 antibiotics-08-00256-t001:** Comparison of mean sum-scores of the ACSS-adjusted ‘typical domain’ symptoms from Day 0 to Day 7 in the (A) microPP and (B) ITT population (Wilcoxon’s 2-sample test).

**A**		**ACSS-Adjusted Symptom Score (microPP)**			
**Day**	**Verum**	**Placebo**	**Difference**	**95% CI**	
Day 0	4.9	5.0	−0.1	[−0.4, +0.4]	*p* = 0.9519
Day 1	4.2	4.8	−0.6	[−1.1, −0.2]	*p* = 0.0086
Day 7	2.1	3.2	−1.1	[−1.6, −0.4]	*p* < 0.0001
**B**		**ACSS-Adjusted Symptom Score (ITT)**			
**Day**	**Verum**	**Placebo**	**Difference**	**95% CI**	
Day 0	4.7	4.9	−0.2	[−0.7, +0.2]	*p* = 0.9519
Day 1	3.9	4.6	−0.7	[−1.3, −0.2]	*p* = 0.0081
Day 7	1.6	3.5	−1.9	[−2.6, −1.1]	*p* = 0.0008

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
