# Peer review of "Effect of a Herbal Therapy on Clinical Symptoms of Acute Lower Uncomplicated Urinary Tract Infections in Women: Secondary Analysis from a Randomized Controlled Trial"

_antibiotics, 2019, doi:10.3390/antibiotics8040256_

Round 1

Reviewer 1 Report

What is the “microPP”, It is needed to define it in content. Suppl. Table1 (Line 476):The numbers are not reasonable, ex, Placebo group. The group is from SAF 98 to microPP 63, but exclusion is 39. ACSS is needed to define in supplementary Table.

Author Response

Comments from reviewer 1:

To the abbreviation microPP an additional explanation has been added in the section Material and Methods and in the Supplementary Material Figure 1. The microPP includes all patients with evaluable microbiological data, i.e. the following patients were excluded:

            - patients whose bacterial count was less than 104 CFU/mL at the time of inclusion

- patients whose urine cultures were contaminated and therefore not evaluable

- patients who took concomitant medication that could influence the results of the urine culture

We have corrected Figure 1 in the supplement. Previously, the two numbers placebo and verum group were reversed with respect to microPP.

Furthermore, the UK English Acute Cystitis Symptom Score (ACSS) was included in the Supplementary Material as Table 3 with the corresponding citation.

Reviewer 2 Report

The manuscript is well written and the data is neatly presented with graphs. The authors show that after reanalysis of the clinical trial data, herbal extracts were superior to placebo and showed symptom relief and prevention of antibiotic use in women with lower uUTI. 

Minor spell checks are required. I recommend this manuscript for publication. 

Author Response

Comments from reviewer 1:

To the abbreviation microPP an additional explanation has been added in the section Material and Methods and in the Supplementary Material Figure 1. The microPP includes all patients with evaluable microbiological data, i.e. the following patients were excluded:

            - patients whose bacterial count was less than 104 CFU/mL at the time of inclusion

- patients whose urine cultures were contaminated and therefore not evaluable

- patients who took concomitant medication that could influence the results of the urine culture

We have corrected Figure 1 in the supplement. Previously, the two numbers placebo and verum group were reversed with respect to microPP.

Furthermore, the UK English Acute Cystitis Symptom Score (ACSS) was included in the Supplementary Material as Table 3 with the corresponding citation.

Comments from reviewer 2:

                We checked the manuscript again for spelling mistakes and corrected them.
We hope we have answered all your comments sufficiently. We are looking forward to your response.
Thank you again for your consideration!
Sincerely,

Reviewer 3 Report

Herbal therapy can be useful in a large number of women who suffer urinary tract infection, and this well designed article proves it.

Author Response

Comments from reviewer 1:

To the abbreviation microPP an additional explanation has been added in the section Material and Methods and in the Supplementary Material Figure 1. The microPP includes all patients with evaluable microbiological data, i.e. the following patients were excluded:

            - patients whose bacterial count was less than 104 CFU/mL at the time of inclusion

- patients whose urine cultures were contaminated and therefore not evaluable

- patients who took concomitant medication that could influence the results of the urine culture

We have corrected Figure 1 in the supplement. Previously, the two numbers placebo and verum group were reversed with respect to microPP.

Furthermore, the UK English Acute Cystitis Symptom Score (ACSS) was included in the Supplementary Material as Table 3 with the corresponding citation.

Comments from reviewer 2:

                We checked the manuscript again for spelling mistakes and corrected them.
